# Identifying and prioritizing potential human-infecting viruses from their genome sequences

**Nardus Mollentze**[1,2]*, **Simon A. Babayan**[2], **Daniel G. Streicker**[1,2]

**1** Medical Research Council-University of Glasgow Centre for Virus Research, Glasgow, United Kingdom,
**2** Institute of Biodiversity, Animal Health and Comparative Medicine, College of Medical, Veterinary and Life Sciences, University of Glasgow, Glasgow, United Kingdom

* nardus.mollentze@glasgow.ac.uk

**Data Availability Statement:** All data and analysis code have been archived on Zenodo (DOI: 10.5281/zenodo.4271479).

**Funding:** D.G.S. and N.M. were supported by a Wellcome Senior Research Fellowship (217221/Z/

## Abstract

Determining which animal viruses may be capable of infecting humans is currently intractable at the time of their discovery, precluding prioritization of high-risk viruses for early investigation and outbreak preparedness. Given the increasing use of genomics in virus discovery and the otherwise sparse knowledge of the biology of newly discovered viruses, we developed machine learning models that identify candidate zoonoses solely using signatures of host range encoded in viral genomes. Within a dataset of 861 viral species with known zoonotic status, our approach outperformed models based on the phylogenetic relatedness of viruses to known human-infecting viruses (area under the receiver operating characteristic curve [AUC] = 0.773), distinguishing high-risk viruses within families that contain a minority of human-infecting species and identifying putatively undetected or so far unrealized zoonoses. Analyses of the underpinnings of model predictions suggested the existence of generalizable features of viral genomes that are independent of virus taxonomic relationships and that may preadapt viruses to infect humans. Our model reduced a second set of 645 animal-associated viruses that were excluded from training to 272 high and 41 very high-risk candidate zoonoses and showed significantly elevated predicted zoonotic risk in viruses from nonhuman primates, but not other mammalian or avian host groups. A second application showed that our models could have identified Severe Acute Respiratory Syndrome Coronavirus 2 (SARS-CoV-2) as a relatively high-risk coronavirus strain and that this prediction required no prior knowledge of zoonotic Severe Acute Respiratory Syndrome (SARS)-related coronaviruses. Genome-based zoonotic risk assessment provides a rapid, low-cost approach to enable evidence-driven virus surveillance and increases the feasibility of downstream biological and ecological characterization of viruses.

## Introduction

Most emerging infectious diseases of humans are caused by viruses that originate from other animal species. Identifying these zoonotic threats prior to emergence is a major challenge

19/Z). Additional funding was provided by the Medical Research Council through program grants MC_UU_12014/8 and MC_UU_12014/12. The funders had no role in study design, data collection and analysis, decision to publish, or preparation of the manuscript.

**Competing interests:** The authors have declared that no competing interests exist.

**Abbreviations:** AUC, area under the receiver operating characteristic curve; AUCm, median area under the receiver operating characteristic curve; CI, confidence interval; COVID-19, Coronavirus Disease 2019; GBM, gradient boosted machine; ICTV, International Committee on Taxonomy of Viruses; ISG, interferon-stimulated gene; SARS, Severe Acute Respiratory Syndrome; SARS-CoV-2, Severe Acute Respiratory Syndrome Coronavirus 2; SHAP, SHapley Additive exPlanations; UPGMA, unweighted pair group method with arithmetic mean; ZAP, zinc-finger antiviral protein.

since only a small minority of the estimated 1.67 million animal viruses may infect humans [1–3]. Existing models of human infection risk rely on viral phenotypic information that is unknown for newly discovered viruses (e.g., the diversity of species a virus can infect) or that vary insufficiently to discriminate risk at the virus species or strain level (e.g., replication in the cytoplasm), limiting their predictive value before the virus in question has been characterized [4–6]. Since most viruses are now discovered using untargeted genomic sequencing, often involving many simultaneous discoveries with limited phenotypic data, an ideal approach would quantify the relative risk of human infectivity upon relevant exposure from sequence data alone. By identifying high-risk viruses warranting further investigation, such predictions could alleviate the growing imbalance between the rapid pace of virus discovery and lower throughput field and laboratory research needed to comprehensively evaluate risk.

Current models can identify well-characterized human-infecting viruses from genomic sequences [7,8]. However, by training algorithms on very closely related viruses (i.e., strains of the same species) and potentially omitting secondary characteristics of viral genomes linked to infection capability, such models are less likely to find signals of zoonotic status that generalize across viruses. Consequently, predictions may be highly sensitive to substantial biases in current knowledge of viral diversity [3,9].

Empirical and theoretical evidence suggests that generalizable signals of human infectivity might exist within viral genomes. Viruses associated with broad taxonomic groups of animal reservoirs (e.g., primates versus rodents) can be distinguished using aspects of their genome composition, including dinucleotide, codon, and amino acid biases [10]. Whether such measures of viral genome composition are specific enough to distinguish host range at the species level remains unclear, but their specificity might arise through several commonly hypothesized mechanisms. First, aspects of antiviral immunity that target nucleotide motifs in viral genomes might select for common mutations in diverse human-associated viruses [11,12]. For example, the depletion of CpG dinucleotides in vertebrate-infecting RNA virus genomes may have arisen to evade zinc-finger antiviral protein (ZAP), an interferon-stimulated gene (ISG) that initiates the degradation of CpG-rich RNA molecules [12]. While ZAP occurs widely among vertebrates, increasingly recognized lineage specificity in vertebrate antiviral defenses opens the possibility that analogous, undescribed nucleic acid targeting defenses might be human (or primate) specific [13]. Second, the frequencies of specific codons in virus genomes often resemble those of their reservoir hosts, possibly owing to increased efficiency and/or accuracy of mRNA translation [14]. By driving genome compositional similarity to human-adapted viruses or to the human genome, such processes may preadapt viruses for human infection [15,16]. Finally, even in the absence of mechanisms that exert common selective pressures on divergent viral genomes, the phylogenetic relatedness of viruses could allow prediction of the potential for human infectivity since closely related viruses are generally assumed to share common phenotypes and host range. However, despite being a common rule of thumb for virus risk assessment, to our knowledge, whether evolutionary proximity to viruses with known human infection ability predicts zoonotic status remains untested.

We aimed to develop machine learning models that use features engineered from viral and human genome sequences to predict the probability that any animal-infecting virus will infect humans given biologically relevant exposure (here, zoonotic potential). Using a large dataset of viruses that had previously been assessed for human infection ability based on published reports, we first build machine learning models that assign a probability of human infection based on virus taxonomy and/or phylogenetic relatedness to known human-infecting viruses and contrast these models to alternatives based on hypothesized selective pressures on viral genome composition that favor human infectivity. We then apply the best performing model

to explore patterns in the predicted zoonotic potential of additional virus genomes sampled from a range of species.

## Results

We collected a single representative genome sequence from 861 RNA and DNA virus species spanning 36 viral families that contain animal-infecting species (S1 Fig). We labeled each virus as being capable of infecting humans or not using published reports as ground truth and trained models to classify viruses accordingly. These classifications of human infectivity were obtained by merging 3 previously published datasets that reported data at the virus species level and therefore did not consider potential for variation in host range within virus species [5,9,17]. Importantly, given diagnostic limitations and the likelihood that not all viruses capable of human infection have had opportunities to emerge and be detected, viruses not reported to infect humans may represent unrealized, undocumented, or genuinely nonzoonotic species. Identifying potential or undocumented zoonoses within our data was an a priori goal of our analysis.

We first evaluated whether phylogenetic proximity to human-infecting viruses elevates zoonotic potential. Gradient boosted machine (GBM) classifiers trained on virus taxonomy or the frequency of human-infecting viruses among close relatives identified by sequence similarity searches ("phylogenetic neighborhood," defined using nucleotide BLAST [10]) outperformed chance (median area under the receiver operating characteristic curve [$AUC_m$] = 0.604 and 0.558, respectively), but were no better than manually ranking novel viruses by the proportion of human-infecting viruses in each family ("taxonomy-based heuristic," $AUC_m$ = 0.596, Fig 1A). This indicates that relatedness-based models were not only unable to identify novel zoonoses that are not close relatives of known human-infecting viruses, but were also largely unable to accurately distinguish risk among closely related viruses (S2 Fig). Moreover, the performance of these models depended on the data available for model training, sometimes performing worse than chance, making them highly sensitive to current knowledge of viral diversity.

We next quantified the performance of GBMs trained on genome composition (i.e., codon usage biases, amino acid biases, and dinucleotide biases), calculated either directly from viral genomes ("viral genomic features") or based on the similarity of viral genome composition to that of 3 distinct sets of human gene transcripts ("human similarity features"): ISGs, housekeeping genes, and all other genes. We hypothesized that if viruses need to adapt to either evade innate immune surveillance for foreign nucleic acids or to optimize gene expression in humans, they should resemble ISGs since both tend to be expressed concomitantly in virus-infected cells. We selected 2 additional sets comprising non-ISG housekeeping genes and all remaining genes to explore whether signals were specific to ISGs. GBMs trained using genome composition feature sets performed similarly when tested separately ($AUC_m$ = 0.688 to 0.701) and consistently outperformed models based on relatedness alone (both the taxonomy-based heuristic and machine learning models trained on virus taxonomy or phylogenetic neighborhood, Fig 1A). Combining all 4 genome composition feature sets further improved, and reduced variance in, performance ($AUC_m$ = 0.740, Fig 1A), suggesting that measures of similarity to human transcripts contained information unavailable from viral genomic features alone. In contrast, adding relatedness features to this combined model reduced accuracy ($AUC_m$ = 0.726) and increased variance (Fig 1A). Averaging output probabilities over the best 100 out of 1,000 iterations of training on random test/train splits of the data (a process akin to bagging, using ranking performance on nontarget viruses to select high performing models) further improved the combined genome feature–based model (area under the receiver operating characteristic curve [AUC] = 0.773, Fig 1B).

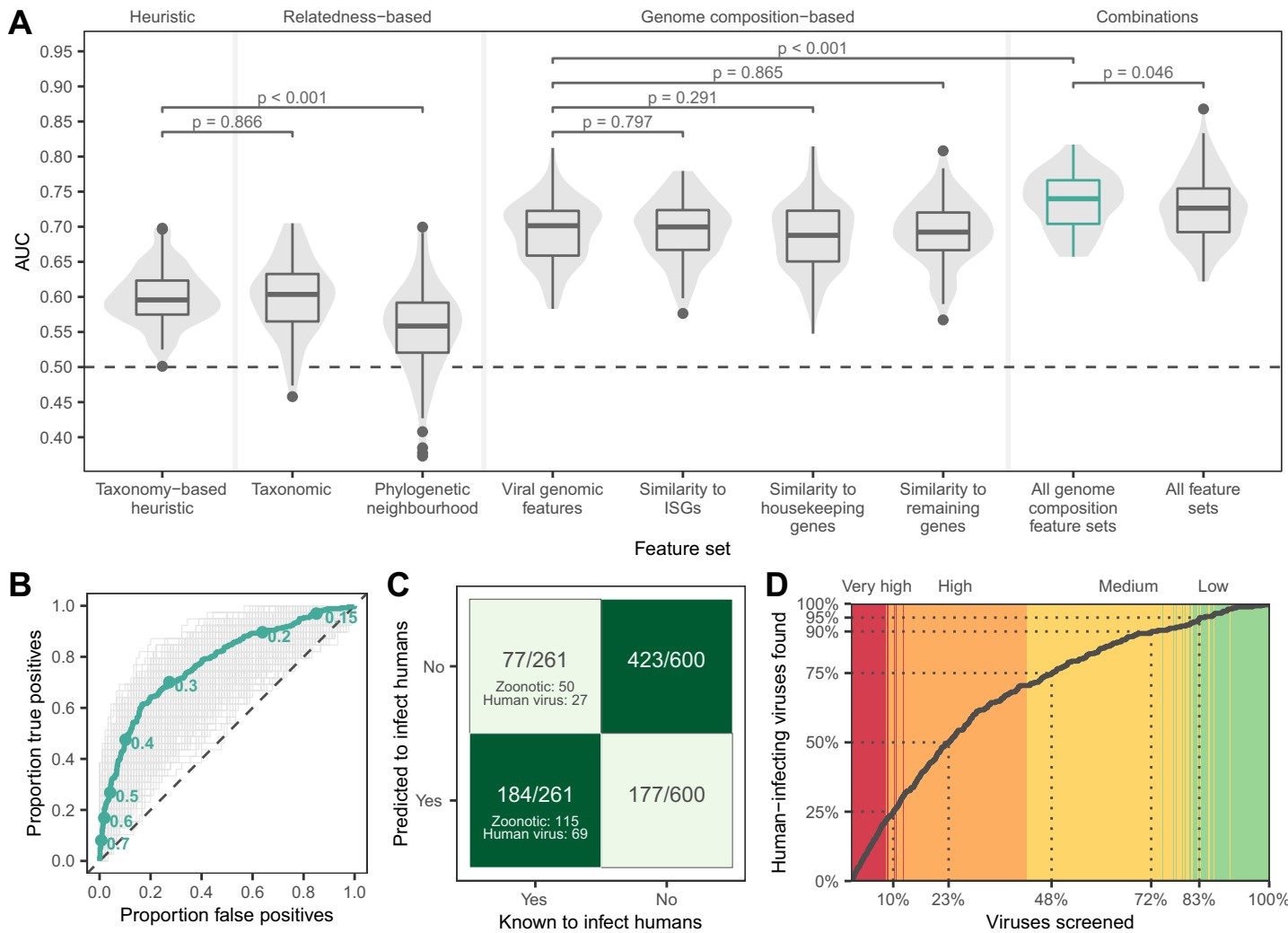

**Fig 1. Machine learning prediction of human infectivity from viral genomes.** (**A**) Violins and boxplots show the distribution of AUC scores across 100 replicate test sets. (**B**) Receiver operating characteristic curves showing the performance of the model trained on all genome composition feature sets across 1,000 iterations (gray) and performance of the bagged model derived from the top 10% of iterations (green). Points indicate discrete probability cutoffs for categorizing viruses as human infecting. (C and D) show binary predictions and discrete zoonotic potential categories from the bagged model, using the cutoff that balanced sensitivity and specificity (0.293). (**C**) Heatmap showing the proportion of predicted viruses in each category. (**D**) Cumulative discovery of human-infecting species when viruses are prioritized for downstream confirmation in the order suggested by the bagged model. Dotted lines highlight the proportion of all viruses in the training and evaluation data that need to be screened to detect a given proportion of known human-infecting viruses. Background color highlights the assigned zoonotic potential categories of individual viruses encountered (red: very high, orange: high, yellow: medium, and green: low). Numerical data underlying this figure can be found at https://github.com/nardus/zoonotic_rank/tree/main/FigureData (doi: 10.5281/zenodo.4271479). AUC, area under the receiver operating characteristic curve.

To estimate model sensitivity and specificity, we converted the mean of predicted probabilities of human infection from the bagged model into binary classifications (i.e., human infecting or not), predicting viruses with predicted probabilities >0.293 as human infecting. This cutoff balanced sensitivity and specificity (both 0.705, Fig 1C), although in principle, higher or lower cutoffs could be selected to prioritize reduction of false positives or false negatives, respectively (Fig 1B). These binary predictions correctly identified 71.9% of viruses that predominately or exclusively infect humans and 69.7% of zoonotic viruses as human infecting, although performance varied among viral families (Fig 1C, S3 Fig). Since binary classifications ignore both the variability between iterations and the rank of viruses relative to each other, we

further converted predicted probabilities of zoonotic potential into 4 zoonotic potential categories, describing the overlap of confidence intervals (CIs) with the 0.293 cutoff from above (low: entire 95% CI of predicted probability ≤ cutoff; medium: mean prediction ≤ cutoff, but CI crosses it; high: mean prediction > cutoff, but CI crosses it; very high: entire CI > cutoff). Under this scheme, the majority (92%) of known human-infecting viruses were predicted to have either medium (21.5%), high (47.1%), or very high (23.4%) zoonotic potential, while only 8% ($N = 21$) had low zoonotic potential (S4 Fig, S1 Table). A total of 18 viruses not currently considered to infect humans by our criteria were predicted to have very high zoonotic potential (S5 Fig), although at least 3 of these (*Aura virus*, *Ndumu virus*, and *Uganda S virus*) have serological evidence of human infection [5,17], suggesting that they may be valid zoonoses rather than model misclassifications. Across the full dataset, 77.2% of viruses predicted to have very high zoonotic potential were known to infect humans (S1 Table). Consequently, studies aimed at confirming human infectivity (e.g., by attempting to infect human-derived cell lines or by serological testing of humans in high-risk populations) while screening viruses in the order suggested by our ranking would have found 23.4% of all known human-infecting viruses in this dataset after screening just the very high zoonotic potential viruses (9.2% of all viruses). More generally, 50% of known human-infecting viruses would have been found after screening the top-ranked 23.3% of viruses and 75% after screening the top 48% of viruses (Fig 1D). In contrast, if relying only on relatedness to known zoonoses, confirming the first 50% of currently known zoonoses would have required screening either 40.2% (taxonomy-based model) or 41.5% (phylogenetic neighborhood–based model) of viruses, a 1.7- to 1.8-fold increase in effort compared to our best model (S6 Fig).

Since genome composition features partly track viral evolutionary history [10], it is conceivable that our models made predictions by reconstructing taxonomy more accurately than the phylogenetic neighborhood estimator or in more detail than available to the taxonomy-based model. We therefore compared dendrograms that clustered viruses by either taxonomy, raw genomic features, or the relative influence of each genomic feature on the model prediction for each virus. The relative influence of each genomic feature on prediction outcomes was measured using the SHapley Additive exPlanations (SHAP) algorithm, which computes the Shapley value for each feature and is increasingly used to improve the interpretability of the decisions made by machine learning models [18]. Shapley values derive from game theory and represent the average marginal contribution of a feature to a prediction across all possible combinations of features [19]. SHAP thus represents complex models as a more interpretable linear combination of values that add up to the final model prediction. As such, SHAP values give a model agnostic measure of how important features are relative to each other when predicting the human infection-ability of a given virus. Here, high levels of similarity in SHAP values between viruses would indicate that they were predicted to have the same human infection status because of the same patterns in their genomic features [20]. Our analysis therefore asked to what extent such similar uses of the same genomic features followed established taxonomic relationships among viruses. While dendrograms using raw feature values closely correlated with virus taxonomy for both human-infecting and other viruses (Baker's [21] γ = 0.617 and 0.492, respectively, $p < 0.001$), dendrograms of SHAP similarity had 10.28- and 2.07-fold reduced correlations with virus taxonomy (γ = 0.060 and 0.238, although this was still more correlated than expected by chance, $p ≤ 0.008$; S7 Fig). Among human-infecting viruses, correlations between SHAP similarity-based clustering and virus taxonomy weakened at deeper taxonomic levels, even though the input genomic features provided sufficient information to partially reconstruct virus taxonomy at the realm, kingdom, and phylum levels (S8 Fig). These results indicate that more taxonomic information was available than was utilized by the trained model to predict human infection ability. Interestingly, dendrograms of SHAP similarity

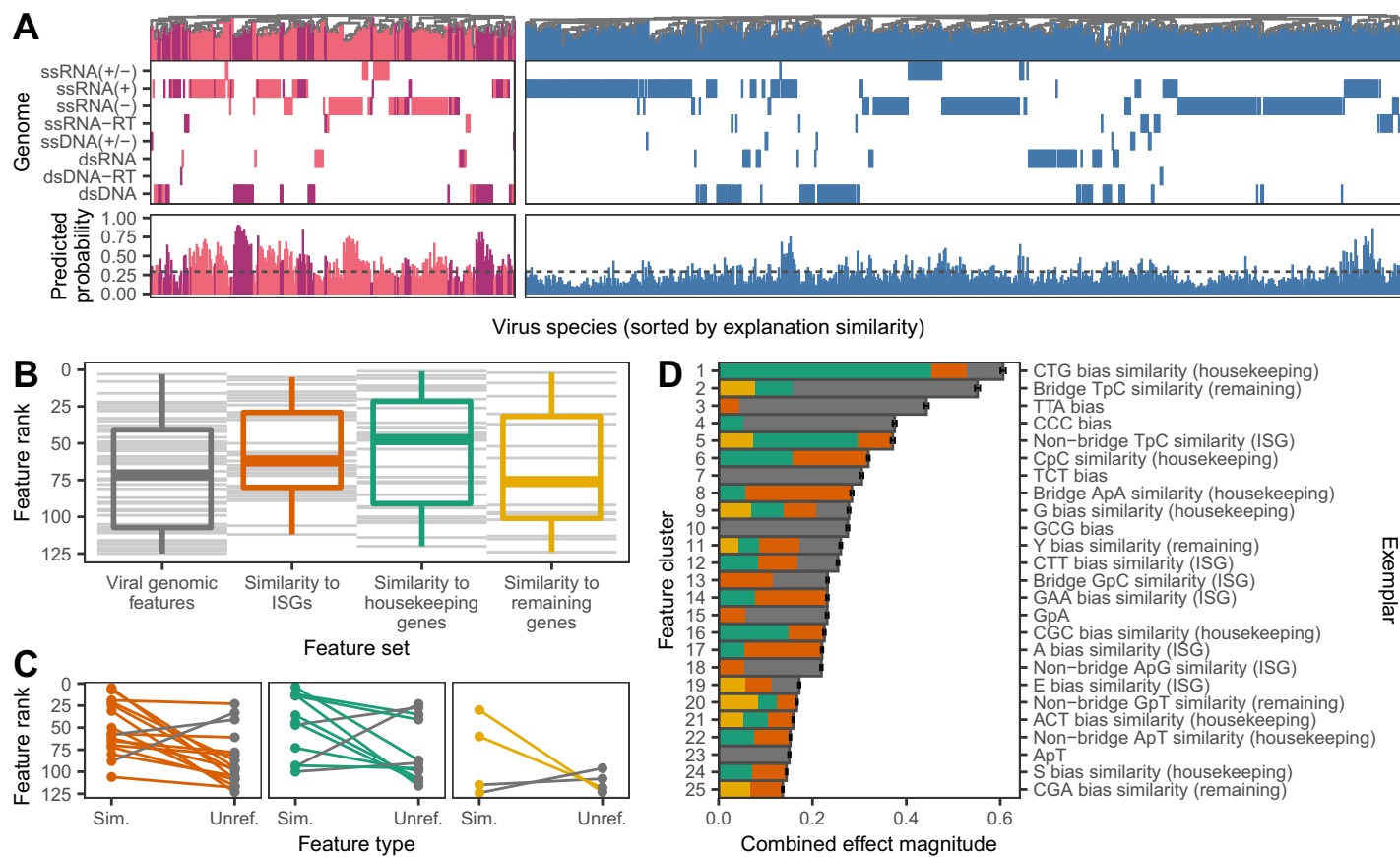

**Fig 2. Genomic determinants of human-infecting viruses. (A)** SHAP value clustering of viruses known to infect humans (primarily human associated, dark purple, and zoonotic, pink) and those with no known history of human infection (blue) shows that similar features predicted human infection across viruses with different genome types (rows). A second set of panels shows the predicted probability of infecting humans for each virus, with the dashed line indicating the cutoff that balances sensitivity and specificity. **(B)** Relative importance of individual features in shaping predictions, determined by ranking features by the mean of absolute SHAP values across all viruses. Gray lines represent individual features; boxplots show the median, 25th/75th percentiles, and range of ranks for each feature set. **(C)** Difference in ranks of features when both unreferenced ("Unref.") and similarity to human genomes ("Sim.") forms were retained in the final model. Lines are colored according to the highest ranked representation in each pairwise comparison; colors as in B. **(D)** Composition of the top 25 most important clusters of correlated features shaping predictions. Discrete clusters of correlated features were identified by affinity propagation clustering. Clusters are shown ranked by the combined effect magnitude of constituent features, defined as the sum of mean absolute SHAP values for all features in the cluster, and the exemplar feature of each cluster is provided on the right axis. Bars represent means (± SEM) across 1,000 iterations and are shaded by the proportion of the cluster from each feature set; colors as in B. Numerical data underlying this figure can be found at https://github.com/nardus/zoonotic_rank/tree/main/FigureData (doi: 10.5281/zenodo.4271479). SHAP, SHapley Additive exPlanations.

showed that even viruses with different genome types—indicating ancient evolutionary divergence or separate origins—clustered together (Fig 2A). Alongside earlier observations on classifier performance (Fig 1A), this suggests that the genome composition-based model outperformed relatedness-based approaches because it found common viral genome features that increase the capacity for human infection across diverse viruses.

Although our analysis was not designed to conclusively identify biological mechanisms underlying genomic predictors of human infection, we nevertheless were able to explore emergent patterns relating to how specific genome composition features and groups of features relate to human infectivity. We first compared the relative influence of features from different genome composition categories (i.e., genomic features versus the 3 sets of human similarity features). Representatives of all genome composition categories were retained in the final model, although we found some evidence that compositional similarity to human

housekeeping genes and ISGs influenced predictions more strongly than unreferenced viral genomic features (Fig 2B and 2C, S1 Text). We next explored the influence of individual features on model predictions in more detail. Unsurprisingly, given that GBMs are designed to make predictions from large numbers of weakly informative features [22], no single feature stood out as the driving force, and many features formed correlated clusters (Fig 2D, S9 Fig). More interestingly, many features had complex, nonlinear relationships with human infection (S10 Fig), such that increased similarity to human gene transcripts did not always increase the likelihood of infecting humans (S1 Text). We speculate that this might reflect trade-offs between different features within viral genomes or context dependencies whereby both mimicry of human transcripts (e.g., for improved translation efficiency) or divergence from human transcripts (e.g., for evasion of nucleotide motif-targeting defenses) may occur for different features (S1 Text).

Finally, we carried out 2 case studies to illustrate the utility of our prediction framework. First, we used the combined genome feature–based model to rank 758 virus species that were not present in our training data. We included all species in the most recent International Committee on Taxonomy of Viruses (ICTV) taxonomy release (#35, April 24, 2020) belonging to animal-infecting virus families and which were originally discovered or sequenced from mammals (including humans), birds, 2 insect orders containing common virus vectors (Diptera and Ixodida), or where the sampled host was not reported. This dataset contained representatives from 38 viral families, including 2 (*Anelloviridae* and *Genomoviridae*), which were not present in data used to train our model. In total, 70.8% of viruses sampled from humans were correctly identified as having either very high ($N = 36$) or high zoonotic potential ($N = 44$; Fig 3A). The remaining human-associated viruses were primarily classified as medium zoonotic potential ($N = 30$), with 3 species predicted to have low zoonotic potential (*Mammalian orthoreovirus* and *Human associated gemykibivirus 2* and *3*; Fig 3A). Within the viral families never previously seen by our model, the majority of human-associated anelloviruses (39/45, 86.6%) were correctly identified as having either very high or high zoonotic potential, consistent with the conclusion that viral genomic features that enhance human infectivity can generalize across viral families. In contrast, all 6 human-associated genomoviruses were classified as either medium or low zoonotic potential. The lower performance on genomoviruses may reflect the unusual genomic structure of this family (circular, single-stranded DNA), which was poorly represented in training (only 2 representatives from the Circoviridae family; S1 Fig) and may impose different selective forces. Further, the small genome sizes of genomoviruses (2.2 to 2.4 kb) may complicate calculation of genomic features due to the low number of nucleotides, dinucleotides, and codons available (cf. S3 Fig). Among the 645 viruses with unknown human infectivity that were sequenced from nonhuman animal or potential vector samples, 45.0% were predicted to have either very high ($N = 41$) or high zoonotic potential ($N = 272$; S11 Fig, S1 Table). The very high zoonotic potential category was dominated by *Papillomaviridae* (34.1%) and *Peribunyaviridae* (19.5%).

We next used a beta regression model to explore how predictions of zoonotic potential varied among host and viral groups. As expected given the performance on our training and evaluation data (Fig 1), the 113 virus species that were sequenced from human samples scored consistently higher than those detected in other hosts ($p < 0.001$; Figs 3A and 4D). Although viruses from putatively high-risk host groups including bats, rodents, and artiodactyls formed a large fraction of our holdout data (with viruses from bats outnumbering even those from humans, S11 Fig), they did not have elevated predicted probabilities of being zoonotic (Fig 4C), and no differences were detected at higher host taxonomic levels (Fig 4A and 4B). This highlights a potential disparity between current sampling efforts for virus discovery/reporting and the distribution of zoonotic risk. In contrast, viruses linked to primates had higher

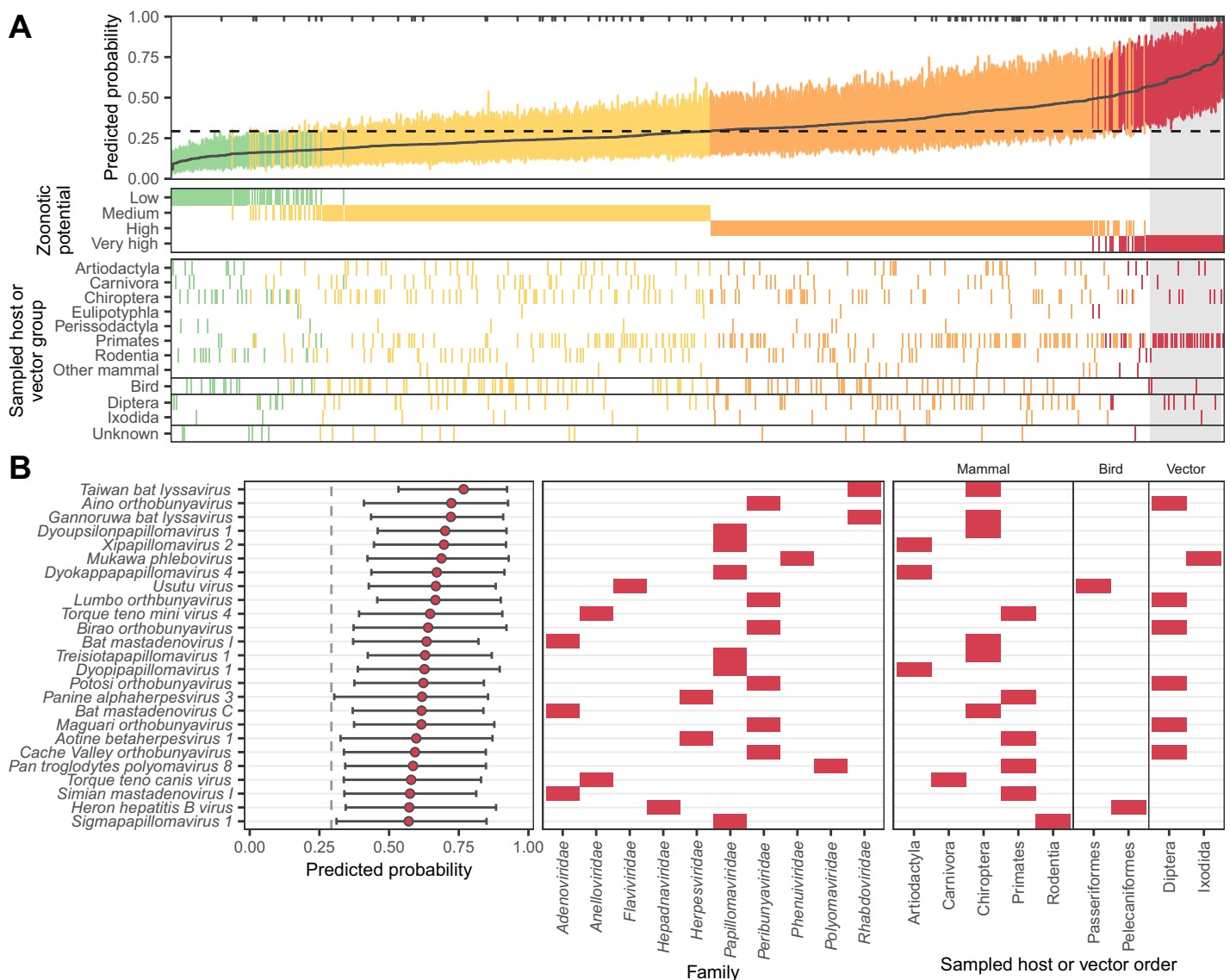

**Fig 3. Probability of human infection predicted from holdout viral genomes. (A)** Predicted probability of human infection for 758 virus species that were not in the training data. Colors show the assigned zoonotic potential categories, with an additional panel showing the host or vector group each virus genome was sampled from. Tick marks along the top edge of the first panel show the location of virus genomes sampled from humans, while a dashed line shows the cutoff that balanced sensitivity and specificity in the training data. The top 25 viruses that were not sampled from humans (contained within the gray box) are illustrated in more detail in **(B)**. Bars show the 95% interquartile range of predicted probabilities across the best performing 10% of iterations (based on the training data), while a solid line **(A)** or circles **(B)** show the mean predicted probability from these iterations. Numerical data underlying this figure can be found in S1 Table and at https://github.com/nardus/zoonotic_rank/tree/main/FigureData (doi: 10.5281/zenodo.4271479).

predicted probabilities of infecting humans, even after accounting for human-associated viruses and the effects of virus family (Figs 3 and 4, S11 Fig). That genome composition-based models predicted elevated zoonotic potential in nonhuman primate–associated viruses despite receiving no information on sampled host further supports host-mediated selective processes as a biological basis for our model's predictions. In addition to relatively rare and small host effects, we observed more pervasive positive and negative effects of virus family on predicted zoonotic status (Fig 4E). Taken together, our results are consistent with the expectation that the relatively close phylogenetic proximity of nonhuman primates may facilitate virus sharing

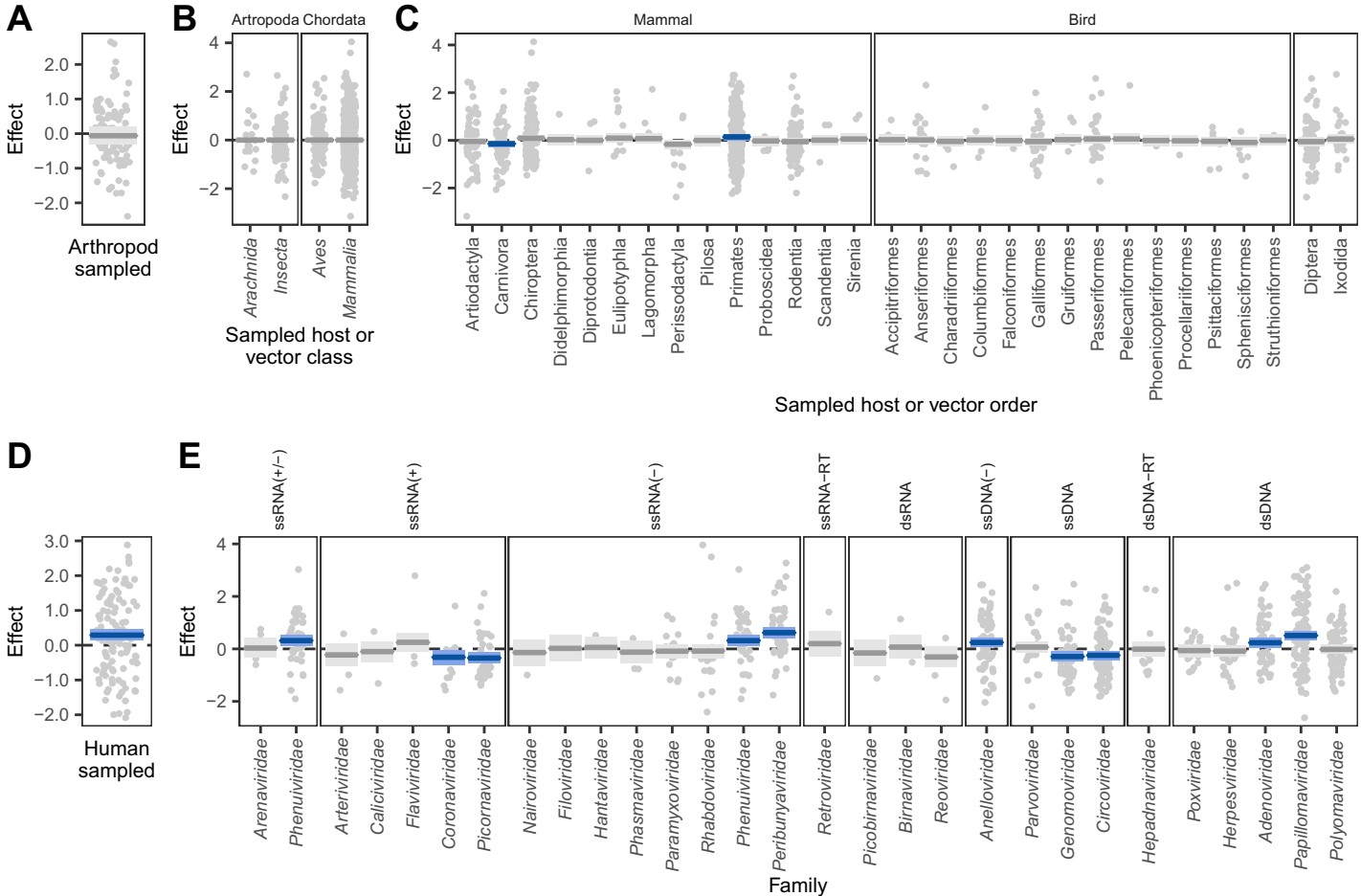

**Fig 4. Factors correlated with the probability of human infection predicted from holdout viral genomes.** Partial effects plots are shown for a beta regression model attempting to explain the mean probability assigned by the bagged model to all viruses in Fig 3A, accounting for whether or not the genome predicted was sequenced from arthropods (as opposed to chordates), **(A)**, random effects for the taxonomic class and order of sampled hosts **(B and C)**, whether the sequence derived from a human sample **(D)**, and a random effect for the virus family represented **(E)**. Points indicate partial residuals, while lines and shaded areas respectively show the maximum likelihood and 95% CI of partial effects. CIs that do not include 0 are highlighted in blue. CI, confidence interval.

with humans and suggest that this may in part reflect common selective pressures on viral genome composition in both humans and nonhuman primates. However, broad differences among other animal groups appear to have less influence on zoonotic potential than virus characteristics [9].

Our second case study used coronaviruses to explore the ability of our combined genome feature–based model to distinguish different virus species within the same family and different genomes within a single virus species. Specifically, we predicted the zoonotic potential of all currently recognized coronavirus species, along with 62 human and animal-derived *Sarbecovirus* genomes all currently classified by the ICTV as *Severe Acute Respiratory Syndrome* (SARS)-*related coronavirus* [23]. All known human-infecting coronaviruses were classified as either medium or high zoonotic potential (Fig 5A). We also identified 2 additional animal-associated coronaviruses—*Alphacoronavirus 1* and the recently described *Sorex araneus coronavirus T14*—as being at least as, or more likely to be capable of infecting humans than known, high-ranking, human-infecting coronaviruses; these should be considered high priority for further research. While this manuscript was in revision, a recombinant

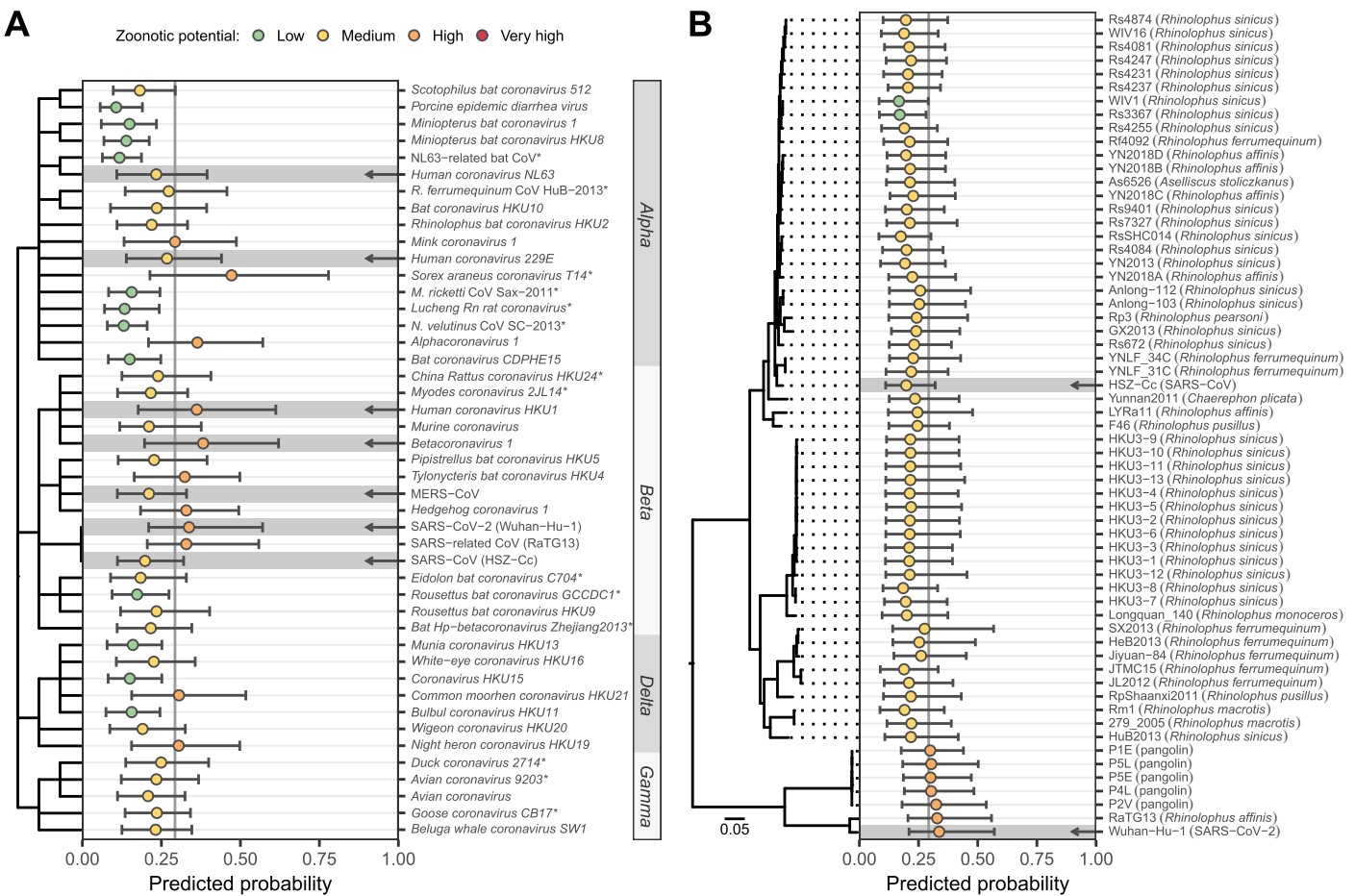

**Fig 5. Probability of human infection predicted from coronavirus genomes. (A)** Predictions for currently recognized *Coronaviridae* species and for 3 variants of SARS-related coronavirus: SARS-CoV (isolate HSZ-Cc, sampled early in the 2003 pandemic), SARS-CoV-2 (isolate Wuhan-Hu-1, sampled early in the current pandemic), and the closely related RaTG13 (sampled from *Rhinolophus affinis* in 2013). A dendrogram illustrates taxonomic relationships, with abbreviated genus names annotated on the right. Arrows highlight known human-infecting species. Asterisks indicate species absent from the training data, also present in Fig 3A. **(B)** Predictions for different representatives of SARS-related coronavirus. The isolation source of animal-associated genomes is indicated in parentheses. A maximum likelihood phylogeny illustrates relationships and was created as described in [6]. The outgroup, BtKy72 (sampled in Kenya in 2007), is not shown. In both panels, bars show the 95% interquartile range of predicted probabilities across the best performing 10% of iterations excluding the species being predicted, while circles show the mean predicted probability from these iterations. Numerical data underlying this figure can be found in S1 Table (panel A) and at https://github.com/nardus/zoonotic_rank/tree/main/FigureData (panel B; doi: 10.5281/zenodo.4271479). MERS-CoV, *Middle East Respiratory Syndrome–related Coronavirus*; *M. ricketti* CoV Sax-2011, *Myotis ricketti alphacoronavirus Sax-2011*; NL63-related bat CoV, *NL63-related bat coronavirus strain BtKYNL63-9b*; *N. velutinus* CoV SC-2013, *Nyctalus velutinus alphacoronavirus SC-2013*; *R. ferrumequinum* CoV HuB-2013, *Rhinolophus ferrumequinum alphacoronavirus HuB-2013*; SARS, Severe Acute Respiratory Syndrome; SARS-CoV-2, Severe Acute Respiratory Syndrome Coronavirus 2.

*Alphacoronavirus 1* was detected in nasopharyngeal swabs from pneumonia patients, further strengthening the case that this species may be zoonotic [24]. We further observed variation in predicted zoonotic potential within coronavirus genera, which was consistent with our current understanding of these viruses. *Alphacoronavirus* and *Betacoronavirus* (the genera that contain known human-infecting species) also contained nonzoonotic species that were correctly predicted to have low zoonotic potential, while the majority of delta- and gammacoronaviruses received relatively low predictions (Fig 5A). These findings further illustrate the capacity of our models to discriminate risk below the virus family or genus levels.

Among sarbecoviruses, most genomes (85.5%) were classified as having medium zoonotic potential, including the causal agent of the 2003 SARS outbreak (Fig 5B). Interestingly,

however, Severe Acute Respiratory Syndrome Coronavirus 2 (SARS-CoV-2; the causative agent of the current Coronavirus Disease 2019 [COVID-19] pandemic), the closely related RaTG13 from a rhinolophid bat, and all 5 closely related pangolin-associated isolates tested were predicted to have high zoonotic potential (although CIs between all sarbecoviruses tested overlapped, Fig 5B). Importantly, these predictions were made using iterations of our model that excluded the 2003 SARS-CoV genome or any other sarbecovirus from training. This finding, together with our observation that relatively few other animal-infecting, allegedly nonzoonotic coronaviruses had similarly high scores, suggests that the elevated risk of SARS-CoV-2 and closely related genomes discovered in animals could have been anticipated via sequencing-based surveillance and might have led to actionable research or surveillance prior to the zoonotic emergence of any sarbecovirus (Fig 5).

## Discussion

In an age of rapid, genomic-based virus discovery, rational prioritization of research and surveillance activities has been an unresolved challenge. While approaches to prioritize known, relatively well-characterized viruses based on a range of common risk factors have been developed [4–6], the large number of viruses still being discovered presents a bottleneck for the very characterization needed to apply such prioritization schemes, necessitating use of expert opinion or surrogate data from related species [6]. Our findings show that the zoonotic potential of viruses can be inferred to a surprisingly large extent from their genome sequence, outperforming current alternatives. Indeed, our results suggest that routine proxies of zoonotic risk that can be applied to poorly characterized viruses including virus taxonomy and relative phylogenetic proximity to human-infecting species [5,9,25] have limited discriminatory power. This has far-reaching implications for how risk is perceived—while it is intuitive to assume that novel viruses that are closely related to known human-infecting viruses are a threat, to our knowledge, this assumption had never been tested. Worryingly, with some training datasets, such relatedness-based models performed worse than random guessing (AUC < 0.5, Fig 1A), suggesting that the current incomplete knowledge of virus diversity could lead to entirely incorrect priorities under such approaches. In contrast, models that exploited features of viral genomes that were at least partly independent of virus taxonomy both generalized predictions across divergent viruses and provided capacity to discriminate risk among closely related virus species.

In requiring only a genome sequence, our approach has quantitative and qualitative advantages over alternative models for zoonotic risk assessment. The most comprehensive alternative model requires virus species-level information on publication count (a proxy for study effort), the diversity of hosts infected, whether or not the virus is vector borne, and the ability to replicate in the cytoplasm [5]. We estimate similar predictive performance for this model ($AUC_m = 0.770$) using a subset of only mammalian viruses (see Methods). However, neither study effort nor knowledge of host range are available for novel viruses, and restricting this model to factors that might reasonably be inferred from virus taxonomy (vector-borne status and ability to replicate in the cytoplasm) performs considerably worse ($AUC_m = 0.647$) than our approach. We were unable to compare the performance of our model to a more recently developed prioritization system based on ecological variables weighted by expert opinion, as metrics of performance were not provided and could not be calculated for a comparable set of viruses with known zoonotic status [6]. Although we emphasize that the viruses included and study objectives differed, that genome-based ranking seems to perform comparably to or better than currently available alternatives that require far more, and often unavailable, data highlights the surprisingly informative signals of human infection ability contained within viral

genomes. Crucially, from the perspective of virus risk assessment, models based purely on genome sequences can be applied much earlier to identify many potential zoonoses immediately after virus discovery and genome sequencing, when data on most other risk factors are still unknown. Ultimately, genome-based rankings could be combined with data on additional known risk factors as they become available [5,6].

By highlighting viruses with the greatest potential to become zoonotic, genome-based ranking allows further ecological and virological characterization to be targeted more effectively. Indeed, studying viruses in the order suggested by genome-based ranking would find many zoonotic viruses much earlier than current taxonomic or phylogenetically informed approaches (Fig 1D, S6 Fig). Nevertheless, we acknowledge that even after applying our models, considerable numbers of viruses may need to undergo confirmatory testing (e.g., infectivity experiments on human-derived cell lines [26]) before significant further research investments, and this need will only increase with ongoing virus discovery. Although these numbers are more manageable considering that experimental validation will be dispersed across virus taxonomic groups that will be studied by different experts, efforts to increase the success rates of virus isolation (a prerequisite for current in vitro validation methods) and to create systems for high-throughput virus host range testing are clearly needed to improve the efficiency of this process [26,27]. Such efforts could further generate valuable feedback data, iteratively improving model performance and consequently reducing the relative proportion of new viruses requiring additional laboratory testing.

Several lines of evidence—including SHAP clustering of viruses with different genome organizations, accurate prediction of human-infecting viruses from families withheld from training, and the prediction of the zoonotic risk of SARS-CoV-2 when withholding data from other zoonotic sarbecoviruses—suggested that our models make predictions using genomic features that predict human infection across divergent virus taxa. From a practical standpoint, this is a major advantage since it means that our model borrows information across families and might therefore anticipate the zoonotic potential of viruses which, due to their rarity or lack of historical precedent, would not otherwise be considered high risk (sometimes referred to as Disease X) [28]. From a broader evolutionary standpoint, the putative existence of convergently evolved features in viral genomes that seem to predispose human infection is a discovery that deserves further mechanistic study. Encouragingly, a substantial literature in vaccine development facilitates genome-wide synonymous recoding of viral genome composition and has established that these changes can dramatically affect viral fitness [29,30]. Our results provide a path by which analogous approaches could test how the features we identified affect viral host range in general and human infection ability in particular. Doing so may reveal novel mechanisms of viral adaptation to humans, which might represent both biologically verified risk factors for the improvement of future models of zoonotic potential and potential therapeutic targets.

We used single exemplar genomes from each virus species to maximize the likelihood of discovering generalizable signatures of human infection while avoiding performance measures that would be overoptimistic for novel viruses. A potential drawback of this approach was that we omitted substantial viral diversity that is not yet formally recognized by the ICTV [3]. However, we contend that including currently unrecognized viruses is unlikely to improve the predictions of our models because (a) most will be nonhuman infecting (an already overrepresented class) and hence provide little additional information; (b) those which do infect humans will not generally be known to do so due to a lack of historic testing, adding misleading signals; and (c) the predictive features identified often span across families, reducing the impacts of taxonomic gaps. The use of single genomes does however mean that the ranks produced here pertain only to the specific genomes tested (in most cases, the NCBI

reference sequence, S1 Table) and may not apply equally across all strains within a species (Fig 5B). We also stress that our model predicts baseline zoonotic potential (i.e., ability to infect humans), which ultimately will be modulated by ecological opportunities for emergence [31,32]. Further, the societal impact of emergence will depend on capacity for human to human spread and on the severity of human disease, which likely require additional nongenomic data to anticipate [31,33].

In summary, we have constructed a genomic model that can retrospectively or prospectively predict the probability that viruses will be able to infect humans. The success of our models required aspects of genome composition calculated both directly from viral genomes and in units of similarity to human transcripts, and some viruses were predicted to be zoonotic due to common genomic traits despite ancient evolutionary divergence. This highlights the potential existence of currently unknown phenotypic consequences of viral genome composition that appear to influence viral host range across divergent viral families. Independently of the mechanisms involved, the performance of our models shows how increasingly ubiquitous and low-cost genome sequence data can inform decisions on virus research and surveillance priorities at the earliest stage of virus discovery with virtually no extra financial or time investment.

## Methods

### Data

Although our primary interest was in zoonotic transmission, we trained models to predict the ability to infect humans in general, reasoning that patterns found in viruses predominately maintained by human-to-human transmission may contain genomic signals that also apply to zoonotic viruses. Data on the ability to infect humans were obtained by merging the data of [5,9,17], which contain species-level records of reported human infections, resulting in a final dataset of 861 virus species from 36 families. In all cases, only viruses detected in humans by either PCR or sequencing were considered to have proven ability to infect humans. All viruses for which no such reports were found were considered to not infect humans (as long as they were assessed for potential human infection by at least one of the studies above), although we emphasize that many of these viruses are poorly characterized and could therefore be unrecognized or unreported zoonoses. We therefore expect our models to further improve as these and new viruses become better characterized. For figures, human-infecting viruses were further separated into primarily human-transmitted viruses and zoonotic viruses, based on virus reservoirs recorded in [9]. Human-infecting viruses for which the reservoir remains unknown were assumed to be zoonotic, while viruses with both a human and nonhuman reservoir cycle (e.g., *Dengue virus*) were recorded as primarily human transmitted, reflecting the primary source of human infection. A representative genome was selected for each virus species, giving preference to sequences from the RefSeq database wherever possible. RefSeq sequences that had annotation issues, represented extensively passaged isolates, or were otherwise not judged to be representative of the naturally circulating virus were replaced with alternative genomes.

### Features

We compared the predictive power of all classifiers to that which could be obtained through knowledge of a virus' taxonomic position alone. This captures the intuitive expectation that viruses can be risk assessed based on knowledge of the human infection abilities of their closest known relatives. To formalize this idea, we first created a simple heuristic that ranks viruses based on the proportion of other viruses in the same family that are known to infect humans ("taxonomy-based heuristic" in Fig 1). Viral family was chosen as the level of comparison because not all viruses are classified in a scheme that includes subfamilies, while lower

taxonomic levels suffer from limited sample size. To further characterize the predictive power of virus taxonomy, we also included potential predictor variables (here termed features) describing virus taxonomy when training classifiers. These included the proportion of human-infecting viruses in each family (calculated from species in the training data only) along with categorical features describing the phylum, subphylum, class, subclass, order, suborder, and family to which each species was assigned ("taxonomic feature set," 8 features). This information was taken from version 2018b of the ICTV master species list (https://talk.ictvonline.org/files/master-species-lists/). To capture taxonomic effects at finer resolution, we summarized the human infection ability of the closest relatives of each virus in the training data (following [10], here termed the "phylogenetic neighborhood feature set," 2 features). To calculate these features, the genome (or genome segments, where applicable) of each virus species was nucleotide BLASTed against a database containing genomes or genome segments for all species in the training data. All BLAST matches with e-value ≤ 0.001 were retained and used to calculate the proportion of human-infecting viruses in the phylogenetic neighborhood of each virus (excluding the current species). We also calculated a "distance-corrected" version of this proportion by reweighting matches according to the proportion of nucleotides $p$ matching the genome of the focal virus:

$$C = \frac{\sum_{i=1}^{N} p_i \cdot x_i}{N},$$

where $N$ is the number of retained BLAST matches for the focal virus species and $x_i = 1$ if matching species $i$ is able to infect humans and equals 0 otherwise. Both the raw and distance-corrected proportion were used in unison to define the phylogenetic neighborhood, allowing classifiers to pick the most informative representation or to combine both pieces of information if needed. In cases where a virus received no matches with e-value ≤ 0.001, both proportions were set to NA to reflect the fact that no information about the phylogenetic neighborhood was available. This occurred for an average of 2% of viruses in each random training and test set (see below).

Various features summarizing the compositional biases in each virus genome were calculated as described in [10]. These included codon usage biases, amino acid biases, dinucleotide biases across the entire genome, dinucleotide biases across coding regions only, dinucleotide biases spanning the bridges between codons (i.e., across base 3 of the preceding codon and base 1 of the current codon), and dinucleotide biases at nonbridge positions ("viral genomic features," 146 features). Similarity features to human RNA transcripts were obtained by first calculating the above compositional biases for human genes. For each gene, the sequence of the canonical transcript was obtained from version 96 of Ensembl [34]. Genes were divided into 3 mutually exclusive sets, encompassing ISGs (taken from [13]; $N = 2,054$), non-ISG housekeeping genes ([35]; $N = 3,172$), and remaining genes ($N = 9,565$). The distribution of observed values for each genome feature was summarized across all genes in a set by calculating an empirical probability density function using version 2.3.1 of the EnvStats library in R version 3.5.1 [36]. The final similarity score for each genome feature of each virus was then calculated by evaluating this density function at the value observed in the virus genome, giving the probability of observing this value among the transcripts of the set of human genes in question (S12 Fig). This yielded 3 feature sets termed "similarity to ISGs," "similarity to housekeeping genes," and "similarity to remaining genes," each containing 146 features.

## Training

Gradient boosted classification trees were trained using the xgboost and caret libraries (versions 0.90 and 6.0–85, respectively) in R [37,38]. Note that while we separate primarily

human-transmitted viruses and zoonotic viruses in some figures, these viruses were considered a single class during training. Thus, models were trained to distinguish viruses known to infect humans from those with no reports of human infection.

A range of models were trained using different combinations of the individual feature sets described above. To reduce runtimes and potential overfitting in subsequent steps, features were subjected to a prescreening step to remove those with little or no predictive value. During this prescreening step, models were trained on a random selection of 70% of the data using all features within the respective feature group (e.g., all ISG similarity features). Training sets were selected using stratified sampling (i.e., selecting positive and negative examples separately) to retain the observed frequencies of positive (known human infecting) and negative (not known to infect humans) virus species. We were unable to additionally stratify training set selection by virus family due to the small numbers of species in many families. All hyperparameters were kept at their default values, except for the number of training rounds, which was fixed at 150. The importance of each feature was summarized across 100 iterations in which the same features were used to train a model on different samples of the full dataset, and the $N$ most predictive features were retained. A range of possible values for $N$ was evaluated by combining all feature sets and using the selected features to optimize and train a final set of model iterations as described below. The final value of $N = 125$ was chosen as the point at which additional features provided no further improvement in performance, measured as the AUC (S13 Fig). Here, AUC measures the probability that a randomly chosen human-infecting virus would be ranked higher than a randomly chosen virus that has not been reported to infect humans. When a given feature set or combination of feature sets comprised <125 features, all features were retained.

Final models were trained using reduced feature sets. To assess the variability in accuracy across different training sets, training was repeated 100 times [10]. In each iteration, training was performed on a random, class-stratified selection of 70% of the available data (here, the training set). Output probabilities were calibrated using half the remaining data (calibration set, again selected randomly and stratified by human infection status), leaving 15% of the full dataset for evaluation of model predictions (test set). In each iteration, hyperparameters were selected using 5-fold cross-validation on the training set, searching across a random grid of 500 hyperparameter combinations. This cross-validation was adaptive, evaluating each parameter combination on a minimum of 3 folds before continuing cross-validation only with the most promising candidates [38]. The parameter combination maximizing AUC across folds was selected and used to train a final model on the entire training set. This model was then used to produce quantitative scores for each species in the calibration and test sets.

Next, outputs were calibrated to allow interpretation as probabilities using the beta calibration method of [39]. A calibration model was fit to scores obtained for the calibration set using version 0.1.0 of the betacal R package. The fitted calibration model was used to produce final output probabilities for virus species in the test set. Finally, to summarize predicted probabilities from the same model trained on different training sets, we averaged the calibrated probabilities across the best performing iterations (a process with similarities to bagging [10]; 1,000 iterations performed). Bagging of virus species from the training data relied on the best 10% of iterations in which each virus occurred in the test set. As such, the focal virus had no influence on the training or calibration of the iterations used in bagging. Further, the performance of each model iteration was recalculated while excluding the focal species from the test set, to prevent accurate prediction of the focal virus from influencing the choice of iterations used for bagging. When predicting the probability of human infectivity for viruses that were completely separated from training (i.e., those in our first case study), we used the best 10% of iterations overall.

Calculating phylogenetic neighborhood features represented a bottleneck in the iterative training strategy described above, because each iteration had a different reference BLAST database, corresponding to the specific training set selected. This would have required repeating BLAST searches at every iteration. To overcome this, a single all-against-all blast search was performed, with search results and e-values subsequently corrected in each iteration to emulate the result that would have been obtained when blasting only against the current training dataset. Specifically, e-values were recalculated as described in [40]:

$$E = \frac{mn}{2^{S'}}$$

where $m$ is the length of the query sequence (in nucleotides), $n$ is the total number of nucleotides in the training set (i.e., the size of the database searched), and $S'$ is bitscore for this particular alignment in the original blast search.

## Feature importance and clustering

To assess the variability in feature importance while accounting for all viruses, feature importance was assessed across all 1,000 iterations produced for bagging above. In each iteration, the influence of features was assessed using SHAP values, an approximation of Shapley values which here describe the change in the predicted log odds of infecting humans attributable to each genome composition feature used in the final model [18]. In each iteration, this produced a SHAP value for each virus–feature combination. The overall importance of each feature was calculated as the mean of absolute SHAP values across all viruses in the training set of a given iteration [20].

Because features tended to be highly correlated, we also report importance values for clusters of correlated features, with the importance of each cluster for individual viruses calculated as the sum of absolute SHAP values across all features in a cluster:

$$I_{c,j,i} = \sum_{f=1}^{N_c} |S_{f,i,j}|,$$

where $I_{c,j,i}$ is the importance of feature cluster $c$ in determining the output score of virus $j$ in iteration $i$, $N_c$ is the number of features in this cluster, and $S_{f,i,j}$ is the SHAP value for feature $f$. The overall importance of each feature cluster in a given iteration was then calculated as the mean of these importance values across all viruses in the training data of that iteration:

$$H_{c,i} = \frac{\sum_{j=1}^{N_j} I_{c,j,i}}{N_j}.$$

Feature clusters were obtained by affinity propagation clustering, which seeks to identify discrete clusters of features centered around a representative feature (the exemplar feature) [41]. Features were clustered using pairwise Spearman correlations as the similarity measure, using version 1.4.8 of the apcluster library in R [42].

To further explore patterns in feature importance across virus species, we followed a strategy similar to [20], clustering viruses based on the average SHAP values assigned to individual features for each virus across all iterations:

$$\bar{S}_{f,j} = \frac{\sum S_{f,i,j}}{N_i}.$$

These values were used to calculate the pairwise Euclidean distances between all virus species using version 2.1.0 of the cluster library in R [43]. Viruses were then clustered using agglomerative hierarchical clustering, calculating distances between clusters as the mean distance between all points in the respective clusters (i.e., unweighted pair group method with arithmetic mean [UPGMA] clustering). To explore patterns common to viruses from each class, clustering was performed separately for known human-infecting and other viruses.

To compare this explanation-based clustering with virus taxonomy, we also constructed a dendrogram based on taxonomic assignments as recorded in version 2018b of the ICTV master species list, using all taxonomic levels from phylum to subgenus. Since some levels of the ICTV taxonomy are not used consistently across all viruses, missing taxonomic levels were interpolated to ensure accurate representation of the underlying taxonomy. For example, for viruses which are not classified in a scheme which includes subfamilies, the next level downstream—genus—was repeated, thereby treating each genus as belonging to a distinct subfamily. Categorical taxonomic assignments were used to calculate pairwise Gower distances between virus species [44], before performing agglomerative hierarchical clustering as described above. We also assessed the ability of underlying genome feature values to reconstruct virus taxonomy by performing hierarchical clustering on a Euclidean distance matrix calculated directly from all genome composition features (i.e., the unreferenced genome, ISG similarity, housekeeping gene similarity, and remaining gene similarity feature sets). The similarity between dendrograms was assessed using the gamma correlation index of [21], as implemented in dendextend version 1.12.0 in R [45]. A null distribution for this statistic was calculated by randomly shuffling the labels (i.e., virus species names) of both dendrograms 1,000 times. To assess the taxonomic depth at which dendrograms were concordant, the Fowlkes–Mallows index was calculated at each possible cut point in the dendrograms being compared [46], again using the dendextend library. As before, a null distribution was generated by randomly shuffling the labels of both dendrograms 1,000 times.

### Ranking holdout viruses

To illustrate the use of our models in practice, the best performing model (i.e., the bagged model trained using the best 125 features selected from among all genome composition-based feature sets, here termed the "combined genome feature–based model") was used to generate predictions for a set of held out viruses. We included all virus species recognized in the latest version of the ICTV taxonomy (release #35, 2019; https://talk.ictvonline.org/taxonomy) that were from families known to contain species that infect animals but which did not occur in our training data because they were absent from the previously described databases of human infection ability used to form the training data [5,9,17]. These included all 36 families represented in the training data, plus *Anelloviridae* and *Genomoviridae*. Names of viruses in the training data were updated to the latest taxonomy and checked for matches in the corresponding ICTV master species list before extracting nonmatching species. For each species, the genome sequence referenced in the ICTV virus metadata resource corresponding to this version of the taxonomy (https://talk.ictvonline.org/taxonomy/vmr) was retrieved and used to calculate genome composition features as described above. The host from which each virus genome was generated was obtained from either the corresponding GenBank entry, the publication first describing the sequence, or the ArboCat database. This host information was used to further subset viruses to include only those sampled from birds, mammals, Diptera (which includes common vectors such as mosquitos and sandflies), and Ixodida (ticks), or for which the sampled host could not be identified. Scoring was performed using each of the top 10% out of 1,000 iterations, as described above, and averaged to obtain the final output probability. CIs

for these mean probabilities were calculated as the 2.5% and 97.5% quantiles of probabilities output by the top 10% of classifiers. Tests for the effects of virus family and sampled host on the predicted probability of infecting humans were performed by fitting a beta regression model using version 1.8–27 of the mgcv library in R. This model fitted the mean predicted probability for each virus as a function of whether a human host was sampled, whether the sample derived from arthropods (both binary fixed effects), random effects for the taxonomic order and class of sampled hosts, and a random effect for virus family. Partial effects plots were generated using code from [9].

## Evaluating existing model performance

To compare the performance of our models to previously published ecological models, we downloaded the fitted models of [5]. The best viral traits model fitted while excluding serological detections (termed "stringent data" in [5], $N = 408$) was then subjected to a testing regime similar to that used for our models. Across 100 iterations, models were refit using a randomly selected subset consisting of 85% of the virus species used in [5]. Each fitted model was then used to predict probabilities of being zoonotic for the remaining 15% of species. This matched the evaluation strategy used for our own models (as reported in Fig 1A), with 85% of the data used during training and calibration, leaving 15% of the data to test model accuracy. Predictions from each iteration were used to calculate AUC using version 1.2.0 of the ModelMetrics package in R.

## Supporting information

**S1 Text. Viral genome compositional predictors of human infection.** Extended discussions of the viral genome features identified by our model as important for prediction of human infection status and potential biological mechanisms underlying the shape of their relationships with human infection status.
(PDF)

**S1 Table. Predicted probabilities of human infection, zoonotic potential categories, and relative priority ranks for all viruses in the manuscript, derived from the combined genome feature–based model.**
(XLSX)

**S1 Fig. Data used in this study for model development and evaluation.** Viruses ($N = 861$ species) predominately transmitted among humans (purple) or from animals to humans (zoonoses, pink) were combined to form the positive class of human-infecting viruses. All other virus species, for which no human infections have been detected (blue), were used as the negative class when training models. Numerical data underlying this figure can be found at https:// github.com/nardus/zoonotic_rank/tree/main/FigureData (doi: 10.5281/zenodo.4271479).
(PDF)

**S2 Fig. Virus ranks produced by relatedness-based models have limited ability to discriminate the zoonotic potential of closely related viruses.** Viruses from the training data are shown ranked by their predicted probability of infecting humans produced by bagged versions of **(A)** the taxonomic feature set-based model and **(B)** the PN-based model. In the top panel of each plot, a solid gray line shows mean bagged probabilities, while colored error bars highlight the region containing 95% of predictions from the iterations used in bagging. A dashed line shows the cutoff that balances sensitivity and specificity. The lower panel in each plot highlights the location of viruses from different families (colored bars) and contains a dark gray background for clarity. Clustering of zoonotic risk by virus family in the taxonomic feature

set-based model (A, lower panel) shows that this model is unable to discriminate high and low risk viral taxa within viral families. This was expected, since this model contained only features resolved at the family level or above. A sequence similarity-based approach was expected to perform better, but while the PN model (B) does show improved resolution, it still fails to separate risk within certain viral families. For example, papillomaviruses and most flaviviruses are ranked as high risk, despite the presence of nonzoonotic species within each family. In contrast, the genome composition model was considerably more accurate (Fig 1, S3 Fig) and was able to assign viruses from the same family into a broader range of risk categories (see S4 Fig), consistent with our current biological understanding that zoonotic ability varies within viral families. Numerical data underlying this figure can be found at https://github.com/nardus/zoonotic_rank/tree/main/FigureData (doi: 10.5281/zenodo.4271479). PN, phylogenetic neighborhood.
(PDF)

**S3 Fig. Family-specific measures of accuracy for the combined genome feature–based model.** AUC here measures the probability of accurately ranking known human-infecting viruses above other viruses from the same family, when ranking viruses using the output from the bagged model based on all genome feature sets. AUC values could not be calculated for families containing <2 human-infecting viruses or <2 viruses not known to infect humans. These families are illustrated in the lower plot in **(A)**, where the y-axis is unitless and overlapping points are stacked, and as gray bars in (B). CIs were calculated using the method of [47,48] and highlight the difficulty of assessing within-family AUC given the relatively small numbers of viruses currently known in most families. We detected no obvious taxonomic pattern in the variation of within-family AUC values. Numerical data underlying this figure can be found at https://github.com/nardus/zoonotic_rank/tree/main/FigureData (doi: 10.5281/zenodo.4271479). AUC, area under the receiver operating characteristic curve; CI, confidence interval.
(PDF)

**S4 Fig. Heterogeneous zoonotic risk predictions for species within viral families.** Viruses in the training data are shown ranked by their mean predicted probability of infecting humans, produced by bagging across iterations of the combined genome feature–based model. In the top panel, a solid gray line shows mean bagged probabilities, while colored error bars highlight the region containing 95% of predictions from the iterations used in bagging. A dashed line shows the cutoff that balances sensitivity and specificity. The lower panel highlights the location of viruses from each family using colored bars and contains a dark gray background for clarity. For a detailed list of viruses and their ranks and priorities, see S1 Table. Model predictions within viral families span risk categories, illustrating the power to discriminate risk at higher taxonomic resolution than models based on conserved features of virus biology (e.g., ability to replicate in cytoplasm or be transmitted by arthropod vectors) or alternative models based on taxonomy or PN (S2 Fig). For the numerical data underlying this figure, see S1 Table. PN, phylogenetic neighborhood.
(PDF)

**S5 Fig. Putative unrecognized zoonoses identified within the training data.** Points and CIs in the left panel show the bagged mean and 95% confidence range on the predicted probability of human infection when using the combined genome feature–based model. Each virus species shown was included in the training data as not currently known to infect humans, but was nevertheless classified in the "very high zoonotic potential" category when included in test sets, which means ≥95% of the bagged iterations predict these viruses as human infecting. The

right panel shows evidence of serological infection for 3 of these viruses (orange), from [5,17]. Absence of serological evidence may reflect lack of diagnostic testing, lack of quantifiable antibody responses, lack of human exposures, or inability to infect humans. For the numerical data underlying this figure, see S1 Table. CI, confidence interval.
(PDF)

**S6 Fig. Cumulative evaluation of human-infecting species when viruses are prioritized for downstream research or surveillance in the order suggested by models trained on different feature sets.** For each feature set (or combination of feature sets), viruses were ranked based on bagged predictions across the top 100 out of 1,000 training iterations. Dotted lines highlight the proportion of all viruses in the training and evaluation data that need to be screened to detect 50% of known human-infecting viruses. Gray lines show the range of accumulation curves expected from random screening in a dataset of this size, simulated by randomly shuffling viruses 1,000 times. Numerical data underlying this figure can be found at https://github.com/nardus/zoonotic_rank/tree/main/FigureData (doi: 10.5281/zenodo.4271479).
(PDF)

**S7 Fig. Concordance and discordance between the virus genome composition features used by machine learning models and virus taxonomy.** Tanglegrams compare hierarchical clustering of viruses by taxonomy (left in all panels) to clustering by **(A and B)** genome features or **(C and D)** model explanations from the combined genome feature–based model (SHAP values). Lines connect individual species in each dendrogram. Terminal branches are colored by family, while connecting lines are colored to indicate human-transmitted (purple), zoonotic (pink), and other viruses (blue). Note that the colors assigned to each family depend on the order of families in the dendrogram and have been optimized for distinguishability of neighboring families, meaning colors do not match across panels. Numerical values used for clustering are displayed in S10 Fig, and can be found at https://github.com/nardus/zoonotic_rank/tree/main/FigureData (doi: 10.5281/zenodo.4271479). SHAP, SHapley Additive exPlanations.
(PDF)

**S8 Fig. Association between virus-specific explanations of model predictions and taxonomy.** Each panel measures clustering similarity (purple/blue line) when cutting virus taxonomy and either feature-based (top row) or SHAP value-based dendrograms (bottom row) into different numbers of clusters. Increasing the number of clusters (k) therefore compares clustering at increasingly shallower parts of the respective dendrograms (i.e., comparing more closely related viruses). A subset of taxonomic levels corresponding to different levels of subdivision is labeled for orientation. A clustering similarity of 1 would indicate complete agreement in the membership of all clusters, while a similarity of 0 indicates no agreement [46]. In all panels, empirical null distributions (gray) were obtained by randomly shuffling the labels of both dendrograms 1,000 times. Clustering similarities significant at the 0.05 level (after Bonferroni correction) are illustrated in purple, while values that are not statistically distinguishable from the empirical null distribution are shown in blue. The correspondence between all dendrograms and virus taxonomy was generally higher than expected by chance. However, clustering of human-infecting viruses shows a lack of correspondence between how genome features are used in the model (as measured using SHAP values, bottom row) and the highest levels of virus taxonomy, even when such information was available among the input features (top row). Specifically, for both human-infecting and nonhuman-infecting viruses, correspondence between clustering and virus taxonomy declines from the family to the realm levels in SHAP-based clustering, but increases in genome feature–based clustering. This indicates that

while genome feature–based clustering closely mirrors high-level virus taxonomy, SHAP-based clustering links divergent viruses (cf. S7 Fig, panel C), implying similar feature usage even among viruses considered unrelated in the taxonomy. Numerical data underlying this figure can be found at https://github.com/nardus/zoonotic_rank/tree/main/FigureData (doi: 10.5281/zenodo.4271479). SHAP, SHapley Additive exPlanations.
(PDF)

**S9 Fig. Clustering reveals high correlation between diverse genome composition feature types.** Discrete clusters of features were obtained using affinity propagation clustering based on the Spearman correlation between all features present in the final model. Clusters are numbered to match their relative importance as defined in Fig 2D. Distances between features are illustrated in 2 dimensions, obtained by multidimensional scaling of the pairwise correlation matrix. Individual clusters are shown to different scales in **(A)** for readability, while **(B)** shows all clusters on the same scale. All points are shown connected to the exemplar feature of that cluster, which is also indicated in bold font. Colors indicate the magnitude of each feature's effect on the combined genome feature–based model's output, calculated as the mean of absolute SHAP values across all viruses in the training data, and averaged across all 1,000 model training iterations. Feature names abbreviated to a single letter indicate amino acid biases, while 3-letter codes written in capital letters indicate codon biases. Dinucleotide biases are abbreviated in the form "CpG" and were calculated separately for codon bridge positions (abbreviation preceded by "b," e.g., "bCpG"), nonbridge positions (preceded by "n," e.g., "nCpG"), and also across all coding sequences of a given genome (no prefix, e.g., "CpG") or across the entire virus genome (suffix ".e," e.g., "CpG.e"). Numerical data underlying this figure can be found at https://github.com/nardus/zoonotic_rank/tree/main/FigureData (doi: 10.5281/zenodo.4271479). SHAP, SHapley Additive exPlanations.
(PDF)

**S10 Fig. Directionality of relationships between genome composition feature values and the estimated odds of infecting humans.** Each panel indicates a discrete cluster of correlated features, numbered by relative importance (see Fig 2D). Points are colored to show the effect of each feature on the predicted log odds that an individual virus infects humans (i.e., the SHAP value for that feature for a given virus, derived from the combined genome feature–based model, and averaged across all 1,000 iterations). The x-axis shows observed feature values (scaled to lie between 0 and 1 to place all features on the same scale), with points jittered to approximate local density where they overlap. Feature abbreviations follow a similar pattern to those in S9 Fig, except that bridge and nonbridge dinucleotide biases are preceded by "br" or "nbr," respectively. Capital letters in parentheses indicate the set of human genes used as baseline to calculate measures of compositional similarity (I = ISG, H = housekeeping genes, and R = remaining genes). Names ending in "(e)" represent features calculated across the entire genome; all other features were calculated with reference to coding sequences only. The exemplar of each cluster is highlighted in bold. Numerical data underlying this figure can be found at https://github.com/nardus/zoonotic_rank/tree/main/FigureData (doi: 10.5281/zenodo.4271479). SHAP, SHapley Additive exPlanations.
(PDF)

**S11 Fig. Distribution of zoonotic potential categories among 758 virus species that were not in the training data. (A)** Distribution of zoonotic potential category assignments by sampled host, for all viruses in Fig 3A. **(B)** Distribution of zoonotic potential categories among viruses from Fig 3A that were sampled from nonhuman hosts, arranged by viral family and genome type. Numerical data underlying this figure can be found at https://github.com/

nardus/zoonotic_rank/tree/main/FigureData (doi: 10.5281/zenodo.4271479).
(PDF)

**S12 Fig. Illustrative calculation of genome feature similarity values.** A histogram (light gray bars) shows the distribution of CTG codon usage bias observed among human non-ISG housekeeping genes. This distribution was used to estimate an empirical probability density function (dark gray line). Evaluating this function for the values of the same feature observed for specific viruses gave the final similarity scores (red dots, with the x-axis representing observed values and the y-axis the new similarity score). Using this similarity score as representing an estimate of how well each virus genome mimics the particular population of human genes resulted in rearrangement of viruses that may help to predict the ability to infect humans. Because similarity scores were calculated via a density function, viruses with feature values significantly outside the range observed for a given set of human genes received a score of 0, but such cases were rare (0.025% of calculated similarity scores, affecting 5% of similarity-related features). Numerical data underlying this figure can be found at https://github.com/nardus/zoonotic_rank/tree/main/FigureData (doi: 10.5281/zenodo.4271479).
(PDF)

**S13 Fig. Ranking performance when training classifiers on restricted numbers of features, with all feature sets included.** Boxplots and shaded areas illustrate the distribution of AUC values obtained when training classifiers on 100 random test:train:calibrate splits. For each set of classifiers, the top *N* most predictive features selected from among all feature sets were retained, with *N* indicated on the x-axis. Numerical data underlying this figure can be found at https://github.com/nardus/zoonotic_rank/tree/main/FigureData (doi: 10.5281/zenodo.4271479). AUC, area under the receiver operating characteristic curve.
(PDF)

## Acknowledgments

We thank Laura Bergner, Richard Orton, Andrew Shaw, Sam Wilson, and David Robertson for helpful discussions and suggestions.

## Author Contributions

**Conceptualization:** Simon A. Babayan, Daniel G. Streicker.

**Data curation:** Nardus Mollentze, Daniel G. Streicker.

**Formal analysis:** Nardus Mollentze.

**Visualization:** Nardus Mollentze.

**Writing – original draft:** Nardus Mollentze.

**Writing – review & editing:** Nardus Mollentze, Simon A. Babayan, Daniel G. Streicker.

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
