## [Editor Report · Decision Letter 0]

15 Dec 2020

Dear Dr. Mollentze, 

Thank you for submitting your manuscript entitled "Identifying and prioritizing potential human-infecting viruses from their genome sequences" for consideration as a Research Article by PLOS Biology.

Your manuscript has now been evaluated by the PLOS Biology editorial staff, as well as by an academic editor with relevant expertise, and I am writing to let you know that we would like to send your submission out for external peer review.

Please re-submit your manuscript within two working days, i.e. by Dec 17 2020 11:59PM.

Kind regards,

Paula

---

Associate Editor

PLOS Biology

---

## [Decision Letter · Decision Letter 1]

3 Mar 2021

Dear Dr. Mollentze,

Thank you very much for submitting your manuscript "Identifying and prioritizing potential human-infecting viruses from their genome sequences" for consideration as a Research Article at PLOS Biology. Your manuscript has been evaluated by the PLOS Biology editors, an Academic Editor with relevant expertise, and by several independent reviewers. I apologize for the time you had to wait meanwhile your manuscript was on review. 

You will see that both reviewers think that this is an interesting study but they have several concerns that need to be solved. In particular, reviewer #2 thinks that you should make the methods more accessible for non-experts and clearly state the advance of their manuscript with respect to previous work, indicating whether 70% of virus categorized correctly is a good result. Reviewer #1 questions the usefulness of the model, and have several questions about your methods. Please address all the reviewers concerns.

In light of the reviews (below), we will not be able to accept the current version of the manuscript, but we would welcome re-submission of a much-revised version that takes into account the reviewers' comments. We cannot make any decision about publication until we have seen the revised manuscript and your response to the reviewers' comments. Your revised manuscript is also likely to be sent for further evaluation by the reviewers. Unfortunately, depending on the revisions, we may need to add a reviewer with AI/ML expertise, which was difficult to secure in this round, but in order to avoid additional delays we have decided to make a decision now.

We expect to receive your revised manuscript within 3 months. 

**IMPORTANT - SUBMITTING YOUR REVISION**

*Re-submission Checklist*

*Published Peer Review*

*PLOS Data Policy*

*Blot and Gel Data Policy*

Sincerely,

Paula

---

Associate Editor,

pjaureguionieva@plos.org,

PLOS Biology

REVIEWS:

Reviewer #1: Genomics and evolution of pathogens.

Reviewer #2: Epidemiology and evolution, mathematical models.

Reviewer #1: In "Identifying and prioritizing potential human-infecting viruses from their genome

Sequences" Mollentze et al. use machine learning approaches to examine the potential for using various genomic features of viral genomes to predict whether the virus will be able to infect humans. Although I cannot evaluate the details of the machine learning approach used, the manuscript is well written and overall, the analysis appears sound. 

The main contribution of this study is demonstrating that there is relevant information within viral genomes for predicting the potential of a virus to infect humans, and that this includes additional information beyond that provided by broad-level taxonomy and phylogenetic similarity. There are two primary, potential implications of this: 1) perhaps viral genomes could be used to prioritize new viruses with zoonotic potential and 2) the genome features that are informative about the potential to infect humans could help us to better understand the adaptation of viruses to certain host species. 

The first of these is the primary focus of this manuscript. However, in practice, I question just how useful this model will be for this type of prioritization given that 25% of the 'negative' training set ("No known human infections") is still predicted to be of high or very high priority. Similarly, >25% of human infecting viruses are categorized as low or medium priority. Given the high number of new viruses that are regularly being identified, and assuming that most will not be capable of human infection, it's not clear to me that using the output of this model will significantly improve our ability to prioritize high risk viruses. The results certainly don't seem compatible with the implications of this statement: "The performance of our models, while imperfect, means that many potential zoonoses can be identified immediately after virus discovery and genome sequencing." Say 10% of new viruses characterized from animal hosts have the potential to infect humans (probably an overestimate), then only about ~23% of the viruses selected by the model to be high or very high priority would actually be capable of infecting humans. Sure, the model will identify some, but how can users distinguish between the true and false positives?

I think it would have been more interesting to see a deeper exploration of the genomic features shown to be associated with human infecting viruses. 

Other concerns:

1. The authors clearly describe how they determine which viruses they consider to be able to infect humans, but they don't clearly describe where they obtained the list of 861 viral species used or how human infecting viruses were categorized as primarily human viruses or zoonotic. These aspects of the methods need to be clarified. 

2. For the novel viruses, it is important to demonstrate that these are really new species (and genomes) and not simply re-categorization of existing/known strains. 

3. It is specified that for the novel viruses tested, viruses were only included id they were "from families known to contain species that infect animals." Was this same criterion also used for the 861 species included in the training/testing set? If not, it seems that this is a biased subset to use for such testing. 

4. ~28% of the novel genomes (71/256) belong to Anelloviridae (Torque Teno viruses), and as noted in the manuscript, no viruses from this group were including in the training/testing set. This seems strange to me. These species certainly existed prior to the latest ICTV release.

5. One confusing aspect of the decision to only include one representative virus from each ICTV species is that certain species are composed several different strains, only some of which have been associated with human infections. For example, Betacoronavirus 1 includes the human endemic virus HCoV-OC43 and also several strains only associated with infections of other mammals. Why was this not taken into account when choosing the representative genome for each species? I would also like to see some discussion of the potential impact of this on model predications. 

6. Of the 43 novel viruses predicted to be unlikely to infect humans, have any been shown to cause human infections?

7. 14/19 of the unknown viruses predicted to be in the very high priority were Torque Teno viruses (anelloviruses). In total, 71 of these viruses were included in the novel set. What proportion of these 71 were isolated from humans? What is the distribution of known human-infecting anelloviruses in the various prediction categories. 

8. The conclusion that SARS2 ranked "considerably higher" than SARS1 in the analysis show in Fig 3C seems potentially a bit misleading. The distributions shown are largely overlapping. Is there a significant difference between these distributions? Given the relatively subtle difference between SARS1 and 2, and the fact that several other coronaviruses, not known to cause human infections, are also ranked similarly to SARS2 in Fig 3C, I think it is important that this statement in the abstract is tempered: "…could have identified the exceptional risk of SARS-CoV-2 prior to the emergence of the first SARS-related coronavirus in humans."

9. For Fig. 3A, please provide some indication of from which host species each of the viruses was isolated.

10. For Fig 3C, Need to provide meaning of circles and error bars in legend. 

11. Fig. S3. The meaning of the dark grey is not clear. Please describe. 

12. The analysis shown in Fig. S9 should include the coronaviruses from animals that are most closely related to SARS-CoV-2. For example: RaTG13, CoVZXC21, CoVZC45 and the pangolin-isolated viruses from 2017 and 2019. 

Reviewer #2: Review of Mollentze et al, PLoS Biol, Feb 2021

- - -

I really liked this paper, and I think it could potentially be published in PLoS Biology. The introduction and results read really well. The discussion needs more context of what's been done in the past. The methods are all AI, and this needs a lot more explanation as many readers of PLoS Biology, all emerging disease ecologists, and even most bioinformaticians will not be familiar with these approaches. It's not enough to cite a paper in the methods and say that some algorithmic/statistical approach was taken. The reader needs to know exactly what this approach does so they can put it in context of the paper's biological assumptions and conclusions.

I don't work in AI or machine learning. Please keep this in mind as you read the comments below as some may sound elementary. However, the paper does need to reach a biology audience with no background in AI/ML.

MAJOR POINTS

1. Most important comment to address is how the probability p was calculated (the probability shown in Figure 3, and maybe in Figure 2A). I tried to follow this and didn't have time to go to the source material. Is p the mean across the best-performing 10% of models as in the Fig 3 caption? If this is the right definition, then if you want to categorize a new virus, which model do you choose from these best 10%? Is p the output probablity described on line 334? Line 372 suggests that p comes from a logistic regression approach, but maybe this is a different p.

2. line 111: what are these 100 or 1000 models? Are these different parameterizations of the same statistical learning algorithm, each parameterization corresponding to a different training set? Is it common in machine-learning papers to label these as different "models"? Line 312 says a "range of models" but this seems to refer to the nine "models" or "feature set combinations" in Figure 1A. Is this correct?

3. How many features are there? Figure 1 shows very clearly how good the prediction is when certain feature groups are included or excluded. But, how many features are in the "Similarity to ISGs" feature set, for example? If a virus is 10kb long, do we choose some length maximum, or some number of windows to look at? And then do we take these fragments and find best local alignments to all N=2054 ISGs? Do we take the score from this local alignment as a similarity score and input it into the ML algorithm? Please just tell me using a back-of-the-envelope calculation (if possible) how many ISG features there are for a 13KB influenza A genome. Later in the paper you settle on N=125 features to use, and I had no idea whether the total number of starting features was nine (Figure 1A), one thousand, or one million.

4. What is a SHAP value? In simple language, made understandable to a non-AI moderately quantitative scientist.

5. What is a "per-virus effect size" of a feature? Is this the effect size in Figure 2? If not, please define both of these.

6. Page 6: 70% of viruses categorized correctly. Is this good? Should we be aiming for something like 90% or 95%? Is this normal for AI methods on complex problems like this one? Is it better than what other approaches have achieved? I feel like this point needs some serious discussion. You can't isolate a novel rare virus in the middle of a rain forest and tell the local animal health department that you're 70% sure it would infect humans. If all approaches in the past have yielded 70% predictability, then maybe the current paper is not much of an advance.

7. In general, how much of an advance is this over previous work done in this area. I see that some of this previous work is described in the introduction, but more detail (either in intro or discussion) would be helpful for the reader to understand the need for a better approach. At some point during the 2007-2012 period, many scientific groups were receiving funding for viral discovery, viral chatter, identification of potential pandemic viruses, identification of zoonotic viruses (see Mark Woolhouse, Nathan Wolfe, Peter Daszak, Paul Kellam, Peter Simmonds). Did this work go anywhere? Did all the deep sequencing approaches work for either virus discovery or virus characterization? What were the biggest drawbacks of the goals set out during this period? Were all the nice-looking diagrams in Nature and Science showing different risk levels of zoonotic jumps correctly put together? Did any of these groups have a 70% accurate prediction method for novel viruses? I am not one of the five authors listed above, and I have no interest or stake in the success/failure of these past projects. This is a genuine question.

MINOR POINTS

8. Abstract - "prior to the emergence of" .. SARS1 or SARS2? Maybe it's simpler to say here that your approach "is able to correctly classify SARS-CoV-2 as having zoonotic potential based on genomic data alone"

9. Abstract - "developed machine learning algorithms" .. did the authors develop new algorithms, adapt existing ones, use standard algorithms inside R-packages?

10. Introduction, second para, "Empirical and theoretical lines of evidence suggest such signals might exist [8,9]." - just a couple more sentences of detail here please on what these lines of evidence are.

11. line 105: tested independently? Or tested seperately?

12. line 106: "Combining all genome composition features.." -- I would write "combining both types of genome composition features.."

13. what is the definition of the "priority categories" (lines 115-116)

14. line 163-164: "such that increased similarity to human genomes did not always increase the likelihood of infecting humans" .. this sounds pretty important and interesting. Can you say more here? Do you know what types of similarity do and don't correspond to more likely human infection? Do you know why?

15. lines 180-182. What are these 55 viruses? Sarbecoviruses? Betacoronaviruses? Does this list include both SARS1 and SARS2?

16. line 186, SARS1 or SARS2?

17. Fig S10. Is the x-axis here "CTG codon usage" (i.e. proportion of Leucines coded for by CTG/CUG), or is it some other bias metric?

18. line 317 "class-stratified", i.e. "stratified by virus taxonomic class", yes?

---

## [Decision Letter · Decision Letter 2]

28 Jul 2021

Dear Dr Mollentze,

Thank you for submitting your revised Research Article entitled "Identifying and prioritizing potential human-infecting viruses from their genome sequences" for publication in PLOS Biology. I have now obtained advice from the original reviewers and have discussed their comments with the Academic Editor. 

Based on the reviews, we will probably accept this manuscript for publication, provided you satisfactorily address the remaining points raised by the reviewers. Please also make sure to address the following data and other policy-related requests.

Reviewer #2 wants you to further clarify the methods to make them more accessible to general biologists, as you explain them in the rebuttal letter.

DATA POLICY:

Regardless of the method selected, please ensure that you provide the individual numerical values that underlie the summary data displayed in the following figure panels as they are essential for readers to assess your analysis and to reproduce it: Figures 1ABCD, 2BC, S6, S8, S12, S13.

**Please also ensure that figure legends in your manuscript include information on where the underlying data can be found, and ensure your supplemental data file/s has a legend.**

We expect to receive your revised manuscript within two weeks.

*Published Peer Review History*

*Early Version*

Sincerely,

Paula

---

Associate Editor,

pjaureguionieva@plos.org,

PLOS Biology

Reviewer remarks:

Reviewer #1: I am satisfied with the changes the authors have made to the manuscript.

Reviewer #2: Thanks to the authors for making a number of improvements to the ms. The referee response doc is clear, but the manuscript less so. The numbers of features are now included in the methods which helps in understanding how much data went into the training. I would recommend putting these numbers into the results section as well -- in parentheses, e.g. (N=146) -- so the reader can follow how much data has gone into each fit/model/parameterization.

1. The SHAP value is still barely explained or not explained. This made pages 9 and 10 of the new ms very hard to follow. It also made most of Figure 2 very hard to understand. I can't tell if this is my shortcoming in this area, or if this is just an opaque measure that will be hard for all biologists to understand. Is a paragraph on the SHAP value and some equations/methods appropriate here? Other biologists will be reading this.

Some other points

2. Line 80, exert not assert

3. Line 90, "we first build statistical models that assign a probability of zoonotic occurrence"

4. Line 109, I would remove the work "predictably"

5. Line 289, remove 95%

6. lines 329-332 do not make sense. SARS is the human virus, so you can't call these the 62 human and animal genomes of SARS. Please use individual lineages or sarbecovirus as a classification here.

7. Likewise, "within SARS coronavirus" on line 369 does not make sense. You probably should use lineage or sub-genus here again.

---

## [Editor Report · Decision Letter 3]

10 Aug 2021

Dear Dr Mollentze,

On behalf of my colleagues and the Academic Editor, Jason Ladner, I am pleased to say that we can in principle offer to publish your Research Article "Identifying and prioritizing potential human-infecting viruses from their genome sequences" in PLOS Biology, provided you address any remaining formatting and reporting issues. These will be detailed in an email that will follow this letter and that you will usually receive within 2-3 business days, during which time no action is required from you. Please note that we will not be able to formally accept your manuscript and schedule it for publication until you have made the required changes.

PRESS

Sincerely, 

Paula Jauregui

---

Paula Jauregui, PhD 

Associate Editor 

PLOS Biology
